# Contribution of RND-Type Efflux Pumps in Reduced Susceptibility to Biocides in *Acinetobacter baumannii*

**DOI:** 10.3390/antibiotics11111635

**Published:** 2022-11-16

**Authors:** Christina Meyer, Kai Lucaβen, Stefanie Gerson, Kyriaki Xanthopoulou, Thorsten Wille, Harald Seifert, Paul G. Higgins

**Affiliations:** 1Institute for Medical Microbiology, Immunology and Hygiene, Faculty of Medicine and University Hospital Cologne, University of Cologne, 50935 Cologne, Germany; 2German Center for Infection Research (DZIF), Partner Site Bonn-Cologne, 50935 Cologne, Germany; 3Center for Molecular Medicine Cologne, Faculty of Medicine and University Hospital Cologne, University of Cologne, 50935 Cologne, Germany

**Keywords:** efflux, biocide, disinfectant, RND-type efflux pump, *Acinetobacter baumannii*

## Abstract

Bacterial efflux pumps are among the key mechanisms of resistance against antibiotics and biocides. We investigated whether differential expression levels of the RND-type efflux pumps AdeABC and AdeIJK impacted the susceptibility to commonly used biocides in multidrug-resistant *Acinetobacter baumannii*. Susceptibility testing and time–kill assays of defined laboratory and clinical *A. baumannii* strains with different levels of efflux pump expression were performed after exposure to the biocides benzalkonium chloride, chlorhexidine digluconate, ethanol, glucoprotamin, octenidine dihydrochloride, and triclosan. While the impact of efflux pump expression on susceptibility to the biocides was limited, noticeable differences were found in kill curves, where AdeABC expression correlated with greater survival after exposure to benzalkonium chloride, chlorhexidine digluconate, glucoprotamin, and octenidine dihydrochloride. AdeABC expression levels did not impact kill kinetics with ethanol nor triclosan. In conclusion, these data indicate that the overexpression of the RND-type efflux pumps AdeABC and AdeIJK contributes to the survival of *A. baumannii* when exposed to residual concentrations of biocides.

## 1. Introduction

*Acinetobacter baumannii* is a Gram-negative pathogen that frequently shows resistance against numerous antimicrobial classes, including carbapenems and other β-lactam agents, aminoglycosides, chloramphenicol, fluoroquinolones, and tetracyclines [1]. This poses a real challenge for the treatment of *A. baumannii* infections in the clinical setting, where the pathogen has a propensity to severely affect critically ill or immunocompromised patients, causing ventilator-associated pneumonia, bloodstream infections, meningitis, urinary tract infections, or wound and soft tissue infections [1]. Current treatment options favour antibiotic combination therapy including ampicillin–sulbactam, carbapenems, polymyxins, or tigecycline [2]. However, pan-drug-resistant *A. baumannii* strains have been reported and carbapenem-resistant *A. baumannii* is considered as priority 1 (“critical”) in the WHO priority pathogens list for research and development of new antibiotics [3]. Carbapenem resistance can be mediated by carbapenem-hydrolysing beta-lactamases, polymyxin resistance by lipopolysaccharide modifications, and tigecycline resistance by active efflux [4]. In addition to antibiotic resistance, decreased susceptibility to biocides has been reported in *A. baumannii* and other bacterial species [5,6]. Biocides are essential decontamination tools used in the clinical setting to prevent hospital-acquired infections [7]. Commonly used biocides include the quaternary ammonium compound (QAC) benzalkonium chloride, the bisbiguanide agent chlorhexidine, ethanol, glucoprotamin, the cationic compound octenidine dihydrochloride, and the bisphenolic agent triclosan. They are used as antiseptics for human skin, wounds, and mucous membranes, as surface disinfectants, and they are also components of numerous consumer products such as mouthwashes, lotions, and soaps [7,8,9].

Concerns have been raised that the widespread use of biocides may lead to reduced biocide susceptibility and cross-resistance to antibiotics, thus facilitating the selection and spread of multi-drug-resistant (MDR) pathogens [5]. Furthermore, inadequate cleaning or exposure to subinhibitory concentrations of biocides might lead to the persistence of *A. baumannii* in the clinical setting [10,11]. Bacterial efflux pumps are among the key mechanisms in reduced antibiotic and biocide susceptibility [5,12]. Broad substrate efflux pumps can extrude a wide number of unrelated antibiotics as well as biocides out of bacterial cells, contributing to the MDR phenotype [12]. Among Gram-negative bacteria, the most clinically relevant efflux pumps belong to the resistance-nodulation cell-division (RND) family [13]. These tripartite pumps are tightly regulated, and mutations in their regulators can lead to efflux pump overexpression [14]. In *A. baumannii,* the RND pumps AdeABC and AdeIJK play a role in conferring multi-drug resistance, including resistance to aminoglycosides, β-lactams, chloramphenicol, fluoroquinolones, macrolides, and tigecycline [1,15,16,17], and have been associated with the efflux of biocides such as benzalkonium chloride and chlorhexidine in studies involving knockout mutants of the efflux transporters AdeB or AdeJ [18], and with triclosan efflux in a clinical correlation study of reduced triclosan susceptibility and efflux overexpression [19]. AdeABC is regulated by the two-component regulatory system AdeRS [20], while AdeIJK is regulated by the TetR-like repressor AdeN [21]. We have previously shown that mutational hotspots in these regulators affect the expression of the corresponding RND efflux pumps, leading to changes in antimicrobial susceptibility [17,22,23]. However, while susceptibility to biocides has been investigated before, the impact of efflux on killing kinetics of biocides has not been well studied. Therefore, the aim of this study was to investigate the impact of RND efflux pumps in *A. baumannii* on their survival when exposed to different concentrations of commonly used biocides.

## 2. Results

### 2.1. Susceptibility Testing

The MIC results of the tested biocides for laboratory and clinical *A. baumannii* strains with efflux pump regulator mutations and differential efflux pump expression levels are summarised in Table 1.

The knockout of *adeRS* resulting in the lack of *adeB* expression in *A. baumannii* ATCC 19606 led to a 4-fold MIC reduction for chlorhexidine digluconate and 2-fold MIC reductions for the biocides glucoprotamin and octenidine dihydrochloride compared to the wildtype parent. In isolate pair 1, *adeS* mutant MB-5, which overexpresses *adeB*, showed a 2-fold increase in MIC for glucoprotamin and triclosan compared to MB-2. MB-43, an *adeS* mutant from isolate pair 2 that shows increased AdeABC efflux, showed a 2-fold MIC increase for chlorhexidine digluconate, whereas in isolate pair 5, no changes in biocide MICs were observed in the *adeR* mutant strain overexpressing *adeB*.

Laboratory and clinical strains with mutations in *adeN* and associated *adeIJK* overexpression, i.e., laboratory strain ATCC 19606 Δ*adeN* and clinical strains MB-273 and MB-1044 in isolate pairs 3 and 4, respectively, did not show any differences in biocide MICs compared to wildtype.

### 2.2. Time–Kill Assay

Time–kill assays were performed with *A. baumannii* ATCC 19606 wt, *A. baumannii* ATCC 19606 Δ*adeRS*, *A. baumannii* ATCC 19606 Δ*adeN*, and *A. baumannii* isolate pairs 1 and 3 (isolate pairs with the highest expression level for the respective efflux pump in the isogenic mutant) to assess bacterial survival in relation to the exposure time. Biocide concentrations with the biggest difference in killing for strains with differential *adeABC* expression are shown in Figure 1. Additional concentrations are shown in Appendix A.

*A. baumannii* ATCC 19606 Δ*adeRS* showed increased killing when grown in the presence of benzalkonium chloride, chlorhexidine digluconate, glucoprotamin, and octenidine dihydrochloride compared to the wildtype parent (Figure 1 and Appendix A). 

Exposure to benzalkonium chloride at 8 mg/L caused >3-log CFU reduction at all time points in *A. baumannii* ATCC 19606 Δ*adeRS*, while the wildtype showed a ~1-log CFU reduction until 3 h, and substantial regrowth by 24 h. Grown in 8 mg/L of chlorhexidine digluconate, *A. baumannii* ATCC 19606 Δ*adeRS* showed a >3-log reduction in CFU count after 1 h and thereafter, while *A. baumannii* ATCC 19606 wt showed a <1-log CFU reduction after 1 h, and regrowth after 3 h (Figure 1). *A. baumannii* ATCC 19606 Δ*adeRS* also showed increased killing at 4 mg/L of chlorhexidine digluconate (Appendix A). The killing rate of *A. baumannii* ATCC 19606 Δ*adeRS* was higher than the wildtype in 8.5 mg/L glucoprotamin, showing ~4-log reduction at 1 h and beyond (Figure 1), and in 4.2 mg/L glucoprotamin during the first 3 h (Appendix A). *A. baumannii* ATCC 19606 Δ*adeRS* exhibited more killing than wildtype during the first 1 h of exposure to 0.5 mg/L octenidine dihydrochloride (Figure 1). 

Considering the range of the results, the *adeRS* knockout showed similar kill kinetics compared to wildtype upon exposure to ethanol. At higher concentrations of the tested biocides, ≥3-log CFU reductions were observed in both strains but killing rates of the *adeRS* knockout strain were higher after 0.5 h for chlorhexidine digluconate and glucoprotamin (Appendix A). 

In isolate pair 1, the strain MB-5 with *adeABC* overexpression exhibited less killing with benzalkonium chloride, chlorhexidine digluconate, glucoprotamin, and octenidine dihydrochloride than the parental strain MB-2 (Figure 1 and Appendix A). During the first 3 h of exposure to chlorhexidine digluconate at 16 mg/L, *A. baumannii* MB-5 showed up to >3-log-fold less killing compared to MB-2 (Figure 1). Similar results were seen with exposure to benzalkonium chloride at 12 mg/L, chlorhexidine digluconate at 8 mg/L, and glucoprotamin at 12 mg/L and at 8.5 mg/L (Appendix A), although both strains showed substantial regrowth at 24 h. MB-2 also showed more initial killing with exposure to 1 mg/L of octenidine (Figure 1). For other concentrations of biocides, including ethanol, little or no (<1 log-fold) differences were observed (Appendix A).

Biocide concentrations with the biggest difference in killing for strains with *adeIJK* overexpression are shown in Figure 2. Additional concentrations are shown in Appendix A. 

Compared to the wildtype, *A. baumannii* ATCC 19606 Δ*adeN* showed less killing upon exposure to 12 mg/L of benzalkonium chloride, and during the first 3 h of exposure to 8 mg/L of benzalkonium chloride and to 16 mg/L of chlorhexidine digluconate. Further, *A. baumannii* ATCC 19606 Δ*adeN* showed slightly less killing (~1-log-fold difference) with 17 mg/L of glucoprotamin during the first 0.5 h, as well as less killing with 1 mg/L of octenidine, and 98,750 mg/L of ethanol at 3 h (≤1-log CFU reduction vs. ≥3-log reduction in wt). When exposed to other concentrations of these biocides, *A. baumannii* ATCC 19606 Δ*adeN* showed similar killing kinetics to *A. baumannii* ATCC 19606 wt (<1-log-fold difference) (Appendix A). 

In *A. baumannii* isolate pair 3, the *adeIJK*-overexpressing strain MB-273 showed earlier regrowth (after 3 h) compared to MB-271 when exposed to benzalkonium chloride at 16 mg/L (Appendix A). For other concentrations and biocides tested, survival rates between MB-273 and MB-271 were comparable (Figure 2 and Appendix A). 

### 2.3. Dose–Response Assay

Dose–response assays were performed with *A. baumannii* ATCC 19606 wt, Δ*adeRS*, Δ*adeN*, and *A. baumannii* isolate pairs 1 and 3 to assess the contribution of RND pumps to triclosan susceptibility (Appendix A). Triclosan was bacteriostatic up to a concentration of 8 mg/L and at higher concentrations reduced the number of viable cells. Based on these results, the concentrations for the time–kill assays with triclosan were chosen (Appendix A). When exposed to triclosan at 16 and 32 mg/L, *A. baumannii* MB-2 was killed slightly more rapidly than MB-5, and *A. baumannii* ATCC 19606 Δ*adeN* showed slightly decreased killing during the first 1 h, within the limits of reproducibility owing to variability in survival rates. *A. baumannii* MB-273 exhibited less killing than MB-271 after 30 min exposure to triclosan at 32 mg/L. For the other isolate pairs and concentrations tested, killing rates were similar.

## 3. Discussion

Reduced susceptibility of pathogens to biocides has repeatedly been reported and may impede efficacious decontamination in the hospital environment [5,24]. In *A. baumannii* and other Gram-negative bacteria, increased tolerance to biocides defined as an increase in MICs above those typical for a species is mediated by efflux pumps such as the chromosomally encoded RND pumps, or by plasmid-encoded pumps from the small multi-drug resistance (SMR) superfamily, also referred to as ’qac’ pumps as they extrude quaternary ammonium compounds (QACs) [18,25,26]. Broad-substrate RND efflux pumps can extrude both antibiotics and biocides from bacterial cells, raising concerns whether they can confer cross-resistance to different antimicrobial agents [12]. In *A. baumannii*, RND efflux pumps have been shown to cause decreased susceptibility to various antimicrobials [17,27].

Bacterial strains were chosen to include both *A. baumannii* laboratory reference strains and efflux regulator knockout strains, which showed differences in susceptibility to various antibiotics depending on their efflux pump expression level [28,29], and clinical isolates, i.e., isolate pairs 1–5, that were previously shown to efflux antimicrobials through overexpressed RND efflux pumps [17,30]. We sought to determine if these pumps are also capable of reducing susceptibility to the commonly used biocides included in this study. We demonstrated that the expression of AdeABC can affect the susceptibility to commonly used biocides such as chlorhexidine digluconate, glucoprotamin, and octenidine dihydrochloride in a strain-dependent manner when tested using broth microdilution. Furthermore, differential efflux pump expression levels were seen to cause differences in kill kinetics although MICs may not be altered.

We found that in addition to previously reported increased resistance to various antimicrobials, clinical *A. baumannii* isolates overexpressing AdeABC exhibited decreased killing when exposed to benzalkonium chloride, chlorhexidine digluconate, glucoprotamin, and octenidine dihydrochloride in time–kill assays. Laboratory reference and knockout *A. baumannii* strains served to verify these results, as they represent a targeted manipulation of efflux pump regulatory genes. 

Our results confirm that the quaternary ammonium compound benzalkonium chloride and the biguanide chlorhexidine are substrates of efflux pump AdeABC in *A. baumannii*, as was found by Rajamohan et al., in a study from 2010 [18], and in a recent study involving *A. baumannii* ATCC 19606 and efflux pump mutants [31]. Unlike in the latter study, where knockout of *adeB* in *A. baumannii* ATCC 19606 provoked a 4-fold decrease in benzalkonium chloride MIC, in our study, *adeABC* expression levels did not affect MICs, but impacted on benzalkonium chloride kill curves. Alongside RND pumps, chlorhexidine susceptibility in *A. baumannii* is also mediated by efflux pump AmvA from the major facilitator superfamily (MFS) [32], and by the specific chlorhexidine-efflux pump AceI from the proteobacterial antimicrobial compound efflux (PACE) family [31,33]. RND-type efflux pumps further contribute to reduced chlorhexidine and QAC susceptibility in *Pseudomonas aeruginosa* [34,35], *Escherichia coli* [36], and *Serratia marcescens* [37].

Our results indicate that AdeABC can extrude the disinfectant glucoprotamin and the cationic antiseptic octenidine in a strain-dependent manner. Increased tolerance to octenidine was also found to be mediated by efflux in *P. aeruginosa* and *Klebsiella pneumoniae*, where the efflux system SmvA (MFS superfamily) extrudes octenidine [38,39]. To our knowledge, no previous studies regarding the impact of efflux on glucoprotamin susceptibility have been carried out in *A. baumannii*. 

The efflux pump AdeIJK has been reported to efflux benzalkonium chloride and chlorhexidine [18], which may add to the predominant impact of AdeABC [31]. However, in our study, the overexpression of AdeIJK in clinical strains did not impact on their susceptibility to these biocides. Nevertheless, while we did not observe any impact on the MIC, the Δ*adeN* laboratory mutant of *A. baumannii* ATCC 19606 exhibited decreased killing at early time points in time–kill assays with benzalkonium chloride and chlorhexidine digluconate, indicating that AdeIJK expression can confer a survival advantage over strains that do not express the pump, although this observation appears to be strain-dependent. 

Except for decreased killing at a single time point in 19606 Δ*adeN*, susceptibility or survival to ethanol was not impacted by differential expression of AdeABC nor AdeIJK in the strains tested in our study. In a study by Prieto et al., exposure to subinhibitory concentrations of ethanol could lead to overexpression of AdeABC in a strain-specific manner and activated the *adeABC* promoter region in *A. baumannii* ATCC 17978, but not in ATCC 19606 [40]. This may suggest that AdeABC can contribute strain-dependently to the development of ethanol tolerance. In the same study, adaptation to ethanol did not activate the promoter region of *adeIJK*. This is according to our findings, where AdeIJK did not notably mediate ethanol tolerance. 

In *P. aeruginosa* and in *Stenotrophomonas maltophilia*, decreased susceptibility to triclosan has been shown to be mediated by RND-type efflux pumps [41,42]. In *A. baumannii*, efflux pumps that have been associated with decreased triclosan susceptibility are AdeIJK [43], AbeM (MATE superfamily) [44], and AdeABC, as a recent study found *A. baumannii* ATCC 19606 Δ*adeB* exhibiting a 4-fold MIC decrease [31]. Yu et al. further correlated decreased triclosan susceptibility with increased *adeB* expression in clinical *A. baumannii* strains [19]. In our study, in contrast, *adeN* knockout or mutant strains that overexpressed AdeIJK did not show MIC differences compared to their corresponding parental strain. The only difference was a one-dilution MIC increase of triclosan in an isolate that overexpressed AdeABC. 

The triclosan time–kill assays showed high variability, making it difficult to interpret the results. This unstable assay might be due to low triclosan solubility and precipitation effects at higher concentrations. Technical challenges due to precipitation of triclosan have also been reported by others [45,46]. In the dose–response assay though, all isolates showed similar curves at lower concentrations, and there was no big difference seen at higher concentrations despite a higher variability, indicating that efflux pump expression level had little effect on triclosan susceptibility in our assays. 

Although the concentrations of biocides tested in the kill curve studies were considerably lower than recommended for routine use in the clinical setting, the results of this study suggest that efflux may play a role in the survival of *A. baumannii* when exposed to lower or residual concentrations of biocides. Other factors that can contribute to survival of *A. baumannii* to biocide exposure include biofilm formation, decreased outer membrane porin expression, and target site modification [47,48,49]. Residual concentrations may become relevant in practice when biocidal formulations are diluted, for example, through the application of hand antiseptics on wet hands, or when biocides such as benzalkonium chloride are used at lower concentrations, e.g., as preservatives in commercial products [50]. Residual concentrations of biocides are further found in different environments, such as low-level benzalkonium chloride in hospital wastewater [51]. These considerations enhance the significance of proper biocide use at recommended concentrations.

## 4. Materials and Methods

### 4.1. Bacterial Strains

*A. baumannii* strains used in the study are listed in Table 2. The *adeRS* and *adeN* knockouts in *A. baumannii* ATCC 19606 were obtained using markerless mutagenesis [52]. *adeRS* knockouts do not express *adeABC* [28], while the knockout of repressor gene *adeN* led to the overexpression of *adeIJK* [29]. Five sets of clinical *A. baumannii* isolate pairs were also included in this study [17,30]. The isolate pairs were defined as two *A. baumannii* isolates collected from the same patient at two different time points, that were shown to be clonally related but differed in their antimicrobial susceptibility to tigecycline owing to increased efflux. 

### 4.2. Biocides

The biocides used in this study were benzalkonium chloride (Sigma-Aldrich, Steinheim, Germany), chlorhexidine digluconate (Molekula Ltd., Darlington, UK), ethanol (Merck, Darmstadt, Germany), glucoprotamin™ (Incidin™ Plus (26 wt% glucoprotamin), Ecolab, Monheim, Germany), octenidine dihydrochloride (AmBeed, Illinois, IL, USA), and triclosan (Molekula Ltd., Darlington, UK). The biocides were diluted in sterile distilled water, except for triclosan, which was diluted in a 1:1 ratio of acetone and phosphate-buffered saline due to its low water solubility.

### 4.3. Susceptibility Testing

Biocide MICs were determined using serial broth microdilution in cation-adjusted Mueller–Hinton broth (CAMHB) according to the Clinical and Laboratory Standards Institute (CLSI) guidelines [54]. Serial 2-fold dilutions were performed with benzalkonium chloride (range tested: 0.06–32 mg/L; recommended working concentrations: 100–200 mg/L when used as preservative, ≥500 mg/L for surface disinfection), chlorhexidine digluconate (range tested: 0.06–32 mg/L; working concentrations: 500–40,000 mg/L), ethanol (range tested: 770–395,000 mg/L, corresponding to 0.1–49.5 vol%; recommended working concentrations: 474,000–750,500 mg/L, corresponding to 60–95 vol%) [55], glucoprotamin (range tested: 0.26–135 mg/L, corresponding to 0.000098–0.05 vol% of Incidin™ Plus; recommended working concentrations: 1350–8100 mg/L of glucoprotamin, corresponding to 0.5–3 vol% of Incidin™ Plus), octenidine dihydrochloride (range tested: 0.03–16 mg/L; recommended working concentrations: 500–2000 mg/L), and triclosan (range tested: 0.016–8 mg/L; recommended working concentrations: 1000–20,000 mg/L). Experiments were repeated three times independently. 

### 4.4. Kill Kinetics

A time–kill assay was performed with minor modifications according to the National Committee for Clinical Laboratory Standards guideline M26-A [56]. Briefly, 200 µL of an overnight culture of the bacterial strain were added to 10 mL of CAMHB and the bacteria were grown at 37 °C in a shaker-incubator until mid-log phase. Then, 50 µL of the log-phase culture were added to a tube containing 10 mL of CAMHB to achieve a final inoculum of approx. 5 × 10^5^ CFU/mL. The equivalent of biocide needed to reach the desired concentration was added. Different biocide concentrations were tested; 0.25, 0.5, 1, 2 × MIC, except for triclosan, for which higher concentrations were needed to reduce growth. At t = 0, 0.5, 1, 3 and 24 h, a 0.5 mL aliquot of the respective biocide solution was removed, serially diluted in 0.9% NaCl, and 100 µL were plated onto Mueller–Hinton agar. After overnight incubation in air at 37 °C, CFU counts were performed. The lower limit of counting was set at 4 log-fold reduction of the initial number of bacterial cells, corresponding to approx. 5 CFU/plate or 50 CFU/mL. CFU/mL values were calculated, and relative survival plotted using GraphPad Prism 6 (GraphPad Software, Inc., San Diego, CA, USA). All time–kill assay experiments were repeated three times independently, including growth and sterility controls.

### 4.5. Dose–Response Assay with Triclosan

Triclosan was added to tubes containing the bacterial inoculum as described above. The final triclosan concentrations ranged from 0.0625 to 128 mg/L. At t = 0 and 3 h, a 0.5 mL aliquot was removed and treated as described above to determine bacterial counts.

## 5. Conclusions

In conclusion, these data suggest that the expression of RND efflux pumps contributes to the survival of *A. baumannii* in the hospital setting despite the use of biocides which can ultimately kill them. The efflux pumps give the organism a window of opportunity to persist and be transferred to other surfaces. As *A. baumannii* is disseminated via the hands of healthcare workers and via contaminated surfaces and medical equipment [1], effective decontamination through proper biocide use at recommended concentrations is particularly important to prevent the spread of the pathogen.

## Figures and Tables

**Figure 1 antibiotics-11-01635-f001:**
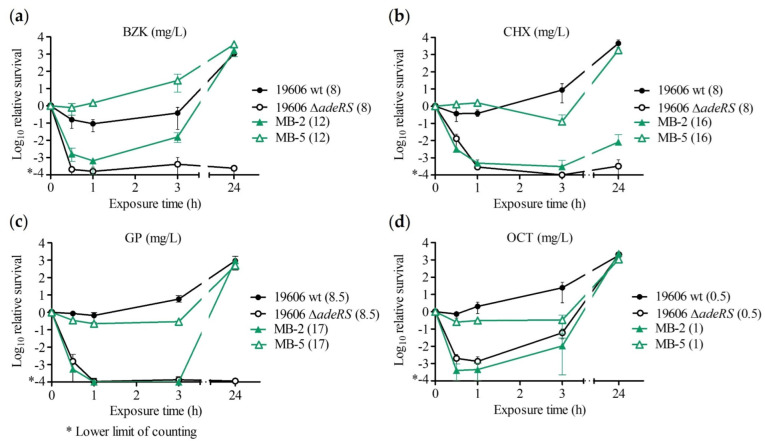
Impact of *adeABC* expression for laboratory strains *A. baumannii* ATCC 19606 and *A. baumannii* 19606 Δ*adeRS* (*adeABC* not expressed) and for clinical strains *A. baumannii* MB-2 and *A. baumannii* MB-5 (*adeABC* overexpressed) in the time–kill assay. Time–kill curves of (**a**) benzalkonium chloride (BZK), (**b**) chlorhexidine digluconate (CHX), (**c**) glucoprotamin (GP), and (**d**) octenidine dihydrochloride (OCT). Tested concentrations (mg/L) written in parentheses. Error bars represent the range from three independent experiments.

**Figure 2 antibiotics-11-01635-f002:**
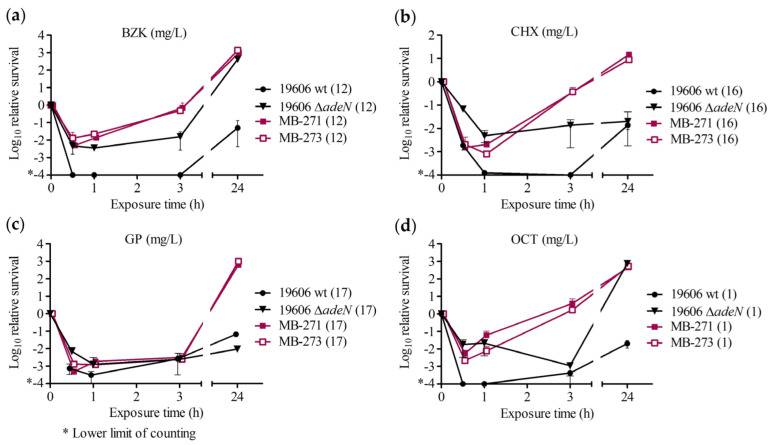
Impact of *adeIJK* overexpression for laboratory strains *A. baumannii* ATCC 19606 and *A. baumannii* 19606 Δ*adeN* (*adeIJK* overexpressed), and clinical strains *A. baumannii* MB-271 and *A. baumannii* MB-273 (*adeIJK* overexpressed) in the time–kill assay. Time–kill curves of (**a**) benzalkonium chloride (BZK), (**b**) chlorhexidine digluconate (CHX), (**c**) glucoprotamin (GP), and (**d**) octenidine dihydrochloride (OCT). Tested concentrations (mg/L) written in parentheses. Error bars represent the range from three independent experiments.

**Table 1 antibiotics-11-01635-t001:** Distribution of biocide MICs determined by microbroth dilution.

	*A. baumannii* Strain	Biocide MIC (mg/L)					
		BZK	CHX	ETH	GP	OCT	TRI
	ATCC 19606 wt	16	16	197,500 *	17	4	1
	ATCC 19606 Δ*adeRS*	16	4	197,500	8.5	2	1
	ATCC 19606 Δ*adeN*	16	16	197,500	17	4	1
Isolate pair 1	MB-2	16	16	197,500	17	4	0.5
	MB-5 (IS*Aba1* disrupts *adeS)*	16	16	197,500	34	4	1
Isolate pair 2	MB-7	16	16	n.d.	17	4	4
	MB-43 (IS*Aba1* disrupts *adeS*)	16	32	n.d.	17	4	4
Isolate pair 3	MB-271	16	16	197,500	34	4	4
	MB-273 (IS*Aba1* disrupts *adeN*)	16	16	197,500	34	4	4
Isolate pair 4	MB-131	32	16	n.d.	17	4	4
	MB-1044 (IS*Aba125* disrupts *adeN*)	32	16	n.d.	17	4	4
Isolate pair 5	Isolate F	16	16	n.d.	17	4	4
	Isolate G (mutation in *adeR*)	16	16	n.d.	17	4	4

BZK: benzalkonium chloride; CHX: chlorhexidine digluconate; ETH: ethanol; GP: glucoprotamin; OCT: octenidine dihydrochloride; TRI: triclosan; n.d.: not determined. * corresponds to 25 vol%.

**Table 2 antibiotics-11-01635-t002:** Bacterial strains used in the current study.

*A. baumannii* Strain			Relevant Characteristics	Reference
Laboratory strains	ATCC 19606		Reference strain	[53]
	ATCC 19606 Δ*adeRS*		*adeABC* not expressed	[28]
	ATCC 19606 Δ*adeN*		2.5-fold increase in *adeIJK* expression	[29]
Clinical strains	Isolate pair 1	MB-2	Wildtype	[17]
		MB-5	IS*Aba1* insertion in *adeS*, 45-fold increase in *adeABC* expression	[17]
	Isolate pair 2	MB-7	Wildtype	[17]
		MB-43	IS*Aba1* insertion in *adeS*, 35-fold increase in *adeABC* expression	[17]
	Isolate pair 3	MB-271	Wildtype	[17]
		MB-273	IS*Aba1* insertion in *adeN*, 6-fold increase in *adeIJK* expression	[17]
	Isolate pair 4	MB-131	Wildtype	[17]
		MB-1044	IS*Aba125* insertion in *adeN*, 2-fold increase in *adeIJK* expression	[17]
	Isolate pair 5	Isolate F	Wildtype	[30]
		Isolate G	Missense mutation in *adeR*, 7-fold increase in *adeABC* expression	[30]

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
