# Peer review of "Contribution of RND-Type Efflux Pumps in Reduced Susceptibility to Biocides in Acinetobacter baumannii"

_antibiotics, 2022, doi:10.3390/antibiotics11111635_

Round 1
Reviewer 1 Report
Comments
1. The virulence factors and disease caused by bacteria need to be added to signifie the study.
2. The effective drugs and their resistance and resistant mechanisms with reference to pump efflux in bacteria need to be described.
3. Other than AdeABC and AdeIJK, the pump's description includes references to antibiotic resistance and biocide resistance.
4. In table 1, the MIC for ETH drugs is greater as compared to other drugs. Justified with reference to its
5. The effects of ethanol in the results are only mentioned in the discussion section.
Author Response
Response to Reviewer 1 Comments
Point 1: The virulence factors and disease caused by bacteria need to be added to signifie the study.
Response 1: We have added a sentence to address the clinical manifestation of the organism (L30-32). Virulence factors are poorly understood and we have not added to this.
Point 2. The effective drugs and their resistance and resistant mechanisms with reference to pump efflux in bacteria need to be described.
Response 2: We have added sentences to address this point. Current treatment options: L33-34, resistance mechanisms to these drugs: L37-39
Point 3. Other than AdeABC and AdeIJK, the pump's description includes references to antibiotic resistance and biocide resistance.
Response 3: I am sorry but we do not understand what is being asked of us.
Point 4. In table 1, the MIC for ETH drugs is greater as compared to other drugs. Justified with reference to its
Response 4: The ETH MIC is listed in the table as mg/L. The value of 197,500 mg/L corresponds to 197.5g of Ethanol in a final volume of 1 litre. This corresponds to 25 vol% of ETH (1L solution of 25 vol% ETH contains 0,25 L of pure ETH. By multiplicating the ETH volume with the ETH density (790,000 mg/L), we obtain the mass of ETH. Thus, the 1 L solution containing 0,25 L of pure ETH contains 0,25 [L] x 790,000 [mg/L] = 197,500 mg of ETH. We thus have 197,500 mg/L of ETH in 25 vol% solution). ETH is used at higher concentrations than other biocides to be effective, therefore its MIC is also higher. The vol% corresponding to the mg/L are also described in the methods section under the point “Susceptibility testing”. We have now added the information that 197,500 mg/L correspond to 25 vol% also below Table 1 to make it clearer.
We also corrected that the in-use concentrations of ETH are 60–95 vol%, instead of 60–99 vol%, as written previously, and added the reference into the Methods section (Reference: [1])
Point 5. The effects of ethanol in the results are only mentioned in the discussion section.
Response 5: The effect of ETH was mentioned in the results section at lines 129 and 158. However, we have also now added another mention of ETH at line 141.
- Boyce, J.M.; Pittet, D. Guideline for hand hygiene in health-care settings. Recommendations of the Healthcare Infection Control Practices Advisory Committee and the HICPAC/SHEA/APIC/IDSA Hand Hygiene Task Force. Society for Healthcare Epidemiology of America/Association for Professionals in Infection Control/Infectious Diseases Society of America. MMWR Recomm Rep 2002, 51, 1-45, quiz CE41-44.
Reviewer 2 Report
Reviewer comments- antibiotics
This manuscript describes “Contribution of RND-type efflux pumps in reduced susceptibility to biocides in Acinetobacter baumannii”. This is a well written article and an interesting study to investigate the impact of RND efflux pumps in A. baumannii on its susceptibility to biocides. There are some major and minor issues in the current manuscript. If authors can address following concerns, this article can be considered for publication.
Major and Minor concerns:
(1) This study needs a clear conclusion. Conclusion section need to be added.
(2) It would be better if authors can add available treatment options for A. baumannii in introduction.
Author Response
Response to Reviewer 2 Comments
Point 1: This study needs a clear conclusion. Conclusion section need to be added.
Response 1: We have added a conclusion to the manuscript (L331-338)
“5. Conclusions
In conclusion, these data suggest that RND efflux pumps contribute to the survival of A. baumannii in the hospital setting despite the use of biocides which can ultimately kill them. The efflux pumps give the organism a window of opportunity to persist and be transferred to other surfaces. As A. baumannii is disseminated via the hands of healthcare workers and via contaminated surfaces and medical equipment [1], effective decontamination through proper biocide use at recommended concentrations is particularly important to prevent the spread of the pathogen.”
Point 2: It would be better if authors can add available treatment options for A. baumannii in introduction.
Response 2: We thank the reviewer for this suggestion, and we have added some treatment options to the manuscript (L33-34).
Reviewer 3 Report
Report
The present article 2019482-- title: Contribution of RND-type efflux pumps in reduced susceptibility to biocides in Acinetobacter baumannii” aimed at investigating the differential expression levels of the RND-type efflux pumps AdeABC and AdeIJK impacted susceptibility to commonly used biocides in multidrug-resistant Acinetobacter baumannii. The aim of this study is important from the medical point of view in terms of the spreading of MDR pathogens and decontamination efficiency in clinical settings.. However, I have certain some comments/suggestions that should be considered before accepting this work for publication, these include:
1. The abstract should be rephrased to highlight the important findings and also a conclusion.
2. Number of the keywords of the abstract should increase to include; Acinetobacter baumannii; RND-type efflux
3. In the introduction (L46-54) should be combined into one paragraph.
4. L54-77, the authors mentioned that “In A. baumannii, the RND pumps AdeABC and AdeIJK play a role in conferring 54 multi-drug resistance, including resistance to aminoglycosides, β-lactams, chloramphenicol, fluoroquinolones, macrolides, and tigecycline [1,25-27], and have been associated 56 with efflux of biocides such as benzalkonium chloride, chlorhexidine and triclosan [28,29]. This means the role of RND has been previously investigated and this will negatively affect the aim of this study. This part needs to be more elaborated and to define the gap that reflects the aim of the current study otherwise this study will lack novelty.
5. The manuscript lacks a conclusion section.
6. In the discussion section, the author should elaborate and discuss in more detail; Why the RND-type efflux pump AdeABC may contribute to the survival of A. baumannii when exposed to residual concentrations of biocides” what are the other factors that could/may contribute to their survival.
Author Response
Response to Reviewer 3 Comments
Point 1. The abstract should be rephrased to highlight the important findings and also a conclusion.
Response 1: We thank the reviewer for this comment, and have amended the abstract to highlight more the conclusions of our work.
Point 2. Number of the keywords of the abstract should increase to include; Acinetobacter baumannii; RND-type efflux
Response 2: We have added the keywords as suggested.
Point 3. In the introduction (L46-54) should be combined into one paragraph.
Response 3: We have merged the two paragraphs as suggested.
Point 4. L54-77, the authors mentioned that “In A. baumannii, the RND pumps AdeABC and AdeIJK play a role in conferring multi-drug resistance, including resistance to aminoglycosides, β-lactams, chloramphenicol, fluoroquinolones, macrolides, and tigecycline [1,25-27], and have been associated with efflux of biocides such as benzalkonium chloride, chlorhexidine and triclosan [28,29]. This means the role of RND has been previously investigated and this will negatively affect the aim of this study. This part needs to be more elaborated and to define the gap that reflects the aim of the current study otherwise this study will lack novelty.
Response 4: We have restructured the concluding paragraph of the introduction to highlight the gaps in the literature with respect to efflux and biocides, to emphasise that we are investigating the killing kinetics as well as susceptibility (L65-7, L71-72, L74).
Point 5. The manuscript lacks a conclusion section.
Response 5: We thank the reviewer for bringing this to our attention, and have added a conclusion to the manuscript (L331-338)
“5. Conclusions
In conclusion, these data suggest that RND efflux pumps contribute to the survival of A. baumannii in the hospital setting despite the use of biocides which can ultimately kill them. The efflux pumps give the organism a window of opportunity to persist and be transferred to other surfaces. As A. baumannii is disseminated via the hands of healthcare workers and via contaminated surfaces and medical equipment [1], effective decontamination through proper biocide use at recommended concentrations is particularly important to prevent the spread of the pathogen.”
Point 6. In the discussion section, the author should elaborate and discuss in more detail; Why the RND-type efflux pump AdeABC may contribute to the survival of A. baumannii when exposed to residual concentrations of biocides” what are the other factors that could/may contribute to their survival.
Response 6: We have added a few sentences to address this at lines 264-266.